



# Ethylamine-Driven Amination of Organic Particles: Mechanistic Insights via Key Intermediates Identification

Peiqi Liu[1], Jigang Gao[1], Yulong Hu[1], Wenhao Yuan[2], Zhongyue Zhou[2], Fei Qi[2], and Meirong Zeng[1]

[1]College of Smart Energy, Shanghai Jiao Tong University, Shanghai 200240, P.R. China

[2]School of Mechanical Engineering, Shanghai Jiao Tong University, Shanghai 200240, P.R. China

*Correspondence to: Meirong Zeng (meirongzeng@sjtu.edu.cn)*

**Abstract:** Atmospheric amines critically contribute to secondary aerosols formation via heterogeneous reactions, yet the molecular mechanisms governing heterogeneous amination chemistry of aerosols remain unclear. Here, we utilize an integrated tandem flow-tube system coupled with online ultrahigh-resolution mass spectrometry to elucidate the amination chemistry of ethylamine (EA) with representative organic aerosol components, including $C_{20}$-$C_{54}$ secondary ozonides (SOZs), $C_{17}$-$C_{27}$ carboxylic acids, and aldehydes. Our experiments provide evidence for the formation of four key intermediates: hydroxyl peroxyamines, amino hydroperoxides, peroxyamines, and amino ethers, which mediate SOZs conversion to hydroxyimines, amides, and imines. Furthermore, dihydroxylamines and hydroxylamines are identified as characteristic intermediates in carboxylic acids and aldehydes amination. Quantitative heterogeneous reactivity measurements ($\Delta\gamma_{eff}$) reveal that SOZs exhibit a pronounced inverse dependence on carbon chain length, e.g., $C_{21}$ SOZ ($\Delta\gamma_{eff} = 1.0 \times 10^{-4}$) > $C_{49}$ SOZ ($\Delta\gamma_{eff} = 5.7 \times 10^{-6}$), with consistently lower reactivity than acids and aldehydes, e.g., $C_{17}$ acid ($\Delta\gamma_{eff} = 2.3 \times 10^{-4}$). The amination mechanism of SOZs is initiated by EA addition, followed by either hydroxyl peroxyamines-mediated dehydration yielding hydroxyimines and amides, or amino hydroperoxides-driven $H_2O_2$ elimination forming imines. For carboxylic acids and aldehydes, EA addition leads to dihydroxylamines and hydroxylamines formation, which subsequently dehydrate to produce amides and imines. These findings provide a mechanistic framework for understanding amine-driven aerosol aging processes that affects atmospheric chemistry, air quality, and climate systems.

## 1 Introduction

Atmospheric aerosols undergo complex chemical transformations that significantly affect human health, environmental quality, and climate systems (Shen et al., 2023; George and Abbatt, 2010). The heterogeneous evolution of organic aerosols, initiated by gaseous amines, drives the formation and growth of nitrogen-containing secondary organic aerosols (SOAs), which are critical components of atmospheric pollution (Na et al., 2007; De Haan et al., 2011; Tian et al., 2024). These transformations are governed by composition-dependent amination mechanisms, with distinct pathways for different organic aerosols, such as carboxylic acids (RCOOHs), aldehydes (RCHOs), and secondary ozonides (SOZs). Quantitative analysis of multiple amination reactions of these particles provides fundamental insights into the chemical evolution process of atmospheric SOAs.

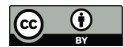



The decomposition of SOZs upon amine exposure initiates through the nucleophilic attack on the carbon atom of SOZs
(Jørgensen and Gross, 2009). However, subsequent reaction pathways remain controversial. Na et al. (2006) demonstrated that
the nucleophilic attack of $NH_3$ on a 3,5-diphenyl-1,2,4-trioxolane (denoted as $C_{14}$ SOZ), derived from the gas-phase ozonolysis
of styrene, induces ring-opening reaction to form a $C_{14}$ amino hydroperoxide. This crucial intermediate subsequently
decompose to yield $H_2O_2$, benzaldehyde, and phenylmethanimine ($C_7H_7N$). Consistently, Almatarneh et al. (2020) and
Jørgensen and Gross (2009) identified $C_2$ amino hydroperoxide intermediates from the reactions of $NH_3$ with a $C_2$ SOZ, derived
from the ozonolysis of ethene. In contrast, Zahardis et al. (2008) observed that the attack of octadecylamine (ODA, $C_{18}H_{39}N$)
on a $C_{18}$ SOZ, produced from the ozonolysis of oleic acid, generates a $C_{36}$ hydroxyl peroxyamine intermediate, ultimately
forming $H_2O$, nonanal, and $C_{27}$ amide. More recently, Qiu et al. (2024) reported that the attack of ethylamine and on a $C_{15}$
SOZ, derived from the ozonolysis of β-caryophyllene, directly open the ring and generates $H_2O$ and a $C_{17}$ amine.
To our knowledge, these key intermediates (amino hydroperoxide and hydroxyl peroxyamine) have not been experimentally
measured in prior studies (Na et al., 2006; Jørgensen and Gross, 2009; Almatarneh et al., 2020; Zahardis et al., 2008), creating
uncertainty about their mechanistic roles in controlling the evolution of SOZ upon amine exposure. The measured stable
amination products (amides, imines, and amines) additionally exhibit inconsistency across studies (Na et al., 2006; Jørgensen
and Gross, 2009; Almatarneh et al., 2020; Zahardis et al., 2008). Furthermore, previous experimental investigations primarily
focused on qualitative products identification, lacking both quantitative reaction rates of SOZ and amine, as well as kinetics
analyses of product formation. These factors limit the evaluation of the amination chemistry in the atmosphere. To mimic the
heterogeneous reactions of SOZs and amine, we generated SOZ particles via the heterogeneous ozonolysis of alkene, their
dominant natural formation pathway (Qiu et al., 2024; Qiu et al., 2022). Specifically, the ozonolysis of squalene (Sqe) was
chosen as model system to generate SOZ particles, building on the demonstration of high SOZ yields (maximum ~ 21% total
yield) from Sqe ozonolysis (Heine et al., 2017). Meanwhile, this Sqe ozonolysis system produces some carbonyl byproducts
(e.g., aldehydes and carboxylic acids) enabling simultaneous quantification of carbonyl aerosols upon amine exposure.
It is widely established that reactions between carboxylic acids and amines typically proceed via acid-base neutralization to
form ammonium salts (Na et al., 2007; Liu et al., 2012; Smith et al., 2010; Gao et al., 2018). However, Ditto et al. (2022)
demonstrated experimentally that the heterogeneous reaction of oleic acid ($C_{18}H_{34}O_2$) and $NH_3$ yields an oleamide ($C_{18}H_{35}ON$)
and $H_2O$. This observation aligns with the calculated pathways by Charville et al. (2011), who suggested that the reaction
between carboxylic acid and amine generates a dihydroxyamine intermediate that subsequently dehydrates to form an amide.
Moreover, significant uncertainties persist regarding the heterogeneous reaction rates of such carboxylic acid and amine
reactions (Fairhurst et al., 2017a; Fairhurst et al., 2017b; Liu et al., 2012). Fairhurst et al. (2017a) reported the heterogeneous
reaction rates (uptake coefficient, γ) ranging from $10^{-1}$ (malonic acid) to $10^{-5}$ (adipic acid) upon ethylamine ($C_2H_7N$) exposure.
They further revealed that the uptake of amines onto low-molecular-weight diacids ($C_3$-$C_8$) is structure-dependent, with higher
γ values observed for odd-carbon diacids than even-carbon ones. Additionally, γ decreases with increasing carbon chain length
of diacids. However, these trends remain unestablished for long-chain acids. Furthermore, to our knowledge, the heterogeneous
uptake coefficients for aldehyde particles upon amine exposure have not been experimentally measured.



Our objective is to investigate the heterogeneous reactions of particulate SOZs, carboxylic acids, and aldehydes upon exposure
to gaseous ethylamine (selected as model amine), using a tandem flowtube reactor. The atmospheric pressure photoionization
high-resolution mass spectrometer (APPI-HRMS) is used to identify reaction products and measure reaction kinetics as a
function of ethylamine exposure. Additionally, the heterogeneous uptake coefficients for SOZs, aldehydes, and acids are
quantified. To interpret the experimental data, the multiphase reaction mechanisms governing the decomposition of SOZs,
carboxylic acids, and aldehydes, as well as the formation of featured amination products are revealed.

## 2 Experimental methods

An integrated experimental system employing a tandem flowtube reactor coupled with APPI-HRMS was developed to examine
the heterogeneous reactions between target organic aerosols upon ethylamine exposure, as illustrated in Fig. 1a. The apparatus
contains three key components: (i) *in-situ* generation of organic particles in the first flowtube reactor from the ozonolysis of
Sqe aerosols, (ii) controlled multiphase reactions between organic particles and ethylamine in the secondary flowtube reactor,
and (iii) online monitoring of chemical compositions of organic particles as a function of ethylamine exposure. To isolate the
contribution of heterogeneous reactions in the secondary flowtube reactor (Fig. 1a), the controlled experiments were conducted
using a single flowtube configuration, as illustrated in Fig. 1b.

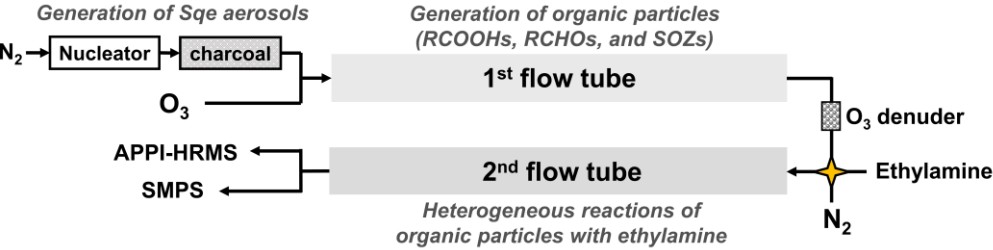

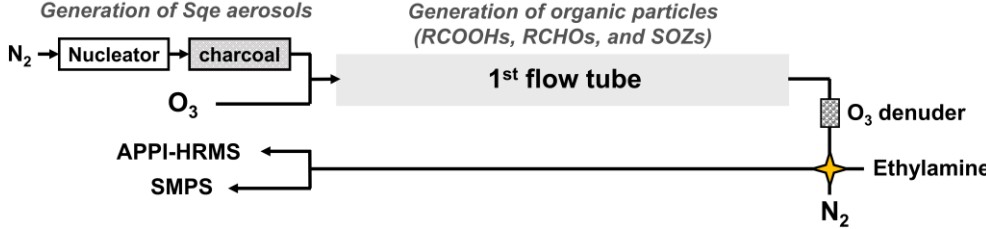


**Figure 1: Schematic diagrams of (a) the tandem flowtube system, and (b) the controlled one flowtube experiment.**
**2.1 Generation of organic particles in the first flowtube reactor**
Polydisperse Sqe aerosols were generated via homogeneous nucleation by passing $N_2$ (300 mL/min) through a Pyrex tube
filled with liquid Sqe (Sigma-Aldrich, 99% purity), located in a tube furnace setting at 145 °C (Fig. 1). Upon exiting the Pyrex



tube, the Sqe vapor cooled and homogeneously nucleated to form aerosols, which were subsequently passed through an annular
activated charcoal denuder to remove any residual gas-phase organics produced in the oven. The average Sqe particle
distribution was log-normal with a mass concentration of 6120 μg/m$^3$ and an average diameter of 209 nm (Fig. S1).
The Sqe aerosol flow was then introduced into the first flowtube reactor (130 cm long and 2.6 cm inner diameter) where they
reacted with O$_3$ to generate organic particles, mainly composed of SOZs, aldehydes, and carboxylic acids (Heine et al., 2017).
O$_3$ was generated by passing 20 mL/min O$_2$ through a corona discharge generator (com-ad-02, Anseros, China), with its
concentration monitored using an O$_3$ analyzer (GM-6000-OEM, Anseros, China). Dilute dry N$_2$ (580 mL/min) was also
introduced into the first flowtube reactor to achieve a total flow rate of 900 mL/min, corresponding to an average residence
time of 46 seconds.
The chemical compositions of organic aerosols generated in the first flowtube reactor were analyzed using APPI-HRMS (Liu
et al., 2024). As illustrated in Figs. S2a and S3a, the major components were identified as SOZs, carboxylic acids, and
aldehydes, consistent with the product distributions measured using vacuum ultraviolet aerosol mass spectrometer (VUV-
AMS) (Heine et al., 2017; Arata et al., 2019). Figure S4 displays the mass signals of representative compounds, including C$_{27}$
aldehyde (C$_{27}$H$_{44}$O), C$_{22}$ acid (C$_{22}$H$_{36}$O$_2$), and C$_{35}$ SOZ (C$_{35}$H$_{58}$O$_3$), as a function of O$_3$ concentration in the first flowtube
reactor. It is demonstrated that, at fixed O$_3$ concentration (e.g., 0.57 ppm), the mass signals of these compounds are stable over
30 min-operation periods with little signal fluctuation. The size distribution of organic particles was monitored using a
Scanning Mobility Particle Sizer (SMPS, TSI 3080L DMA and 3776 CPC), revealing an average diameter of 201 nm (Fig.
S1).

## 2.2 Reactions of organic particles with ethylamine in the secondary flowtube reactor

Upon exiting the first flowtube reactor and subsequent O$_3$ denuder, the organic particles were introduced into a secondary
flowtube reactor (130 cm long and 2.6 cm inner diameter) to investigate their heterogeneous reactions with ethylamine (Fig.
1a). This coupled reactor configuration, combining organic aerosols generated in the first flowtube reactor with amine exposure
in the secondary flowtube, is designated as the tandem 2FT experimental system. Ethylamine was supplied from a standard
gas cylinder (1000 ppm ethylamine balanced with nitrogen; Wetry, Shanghai, China). A mixture of organic particle flow,
ethylamine, and diluent N$_2$ was introduced into the secondary flowtube reactor, maintaining a total flow rate of 1100 mL/min
(corresponding to an average residence time of 37 seconds). The ethylamine concentration in the secondary flowtube reactor
was varied from 0 to 43.45 ppm. All experiments were conducted at atmospheric pressure and room temperature.

## 2.3 Controlled experiments in one flowtube reactor

As widely demonstrated (Bos et al., 2006; Fredenhagen and Kuhnol, 2014), the reaction kinetics measured using the APPI
technique could be influenced by potential photochemical side reactions (Fig. S5). To eliminate contributions from interactions
between ethylamine and organic aerosols in the APPI region, control experiments were designed (Fig. 1b). In these control
experiments, organic aerosols generated in the first flowtube reactor were introduced directly into the APPI region, bypassing





the secondary flowtube reactor. By subtracting the reaction kinetics of the 1FT controlled experiments from those obtained in
the tandem 2FT experiments, the net contribution of heterogeneous reactions occurring in the secondary flowtube reactor were
quantitatively determined (Fig. S6).
**2.4 Real-time detection system and data analysis for heterogeneous reactions**
A portion of the particle stream was sampled by the SMPS to measure particle size distribution and concentration. The
remaining flow (800 mL/min) was directed into the ionization region of the APPI-HRMS (Orbitrap Fusion, Thermo Scientific)
for real-time chemical characterization (Fig. S5). Additional details on the application of APPI-HRMS for quantifying
heterogeneous reactions of particles are available in our previous work (Liu et al., 2024).
By monitoring the mass signals of organic particles (denoted as [Particle]) as a function of ethylamine exposure, defined as
the concentration of ethylamine ([ethylamine]) × residence time ($t$), the heterogeneous decay rate ($k_{particle}$) was determined
through fitting the decay profiles to an exponential function (Equation 1) (Smith et al., 2009; Liu et al., 2024). The effective
uptake coefficient ($\gamma_{eff}$), representing the probability of reactive particle decay upon ethylamine collisions, was then calculated
using Equation 2 (Liu et al., 2024; Smith et al., 2009). In Equation 2, $D$, $\rho_0$, $N_A$, $\bar{c}$, and M correspond to particle diameter,
density, Avogadro's number, mean speed of ethylamine, and molar mass of reactant molecules, respectively. In this work, the
heterogeneous reaction rates measured in the 2FT flowtube reactor experiments and single flowtube reactor (1FT) experiments
were designated as $\gamma_{eff,\ 2FT}$ and $\gamma_{eff,\ 1FT}$, respectively. The net contribution of heterogeneous reactions in the secondary flowtube
reactor was quantified by subtracting $\gamma_{eff,\ 1FT}$ from $\gamma_{eff,\ 2FT}$, yielding the differential uptake coefficient ($\Delta\gamma_{eff}$) as defined in
Equation 3. Figure S6 presents the decay kinetics and corresponding $\Delta\gamma_{eff}$ values for representative compounds: $C_{27}$ aldehyde,
$C_{27}$ acid, and $C_{20}$ SOZ. For instance, $\gamma_{eff,\ 2FT}$ and $\gamma_{eff,\ 1FT}$ values for $C_{20}$ SOZ were determined to be $8.0 \times 10^{-5}$ and $6.2 \times 10^{-5}$,
respectively, resulting in $\Delta\gamma_{eff} = 1.8 \times 10^{-5}$.
$$\frac{[Particle]}{[Particle]_0} = \exp\left(-k_{particle} \times [Ethylamine] \times t\right) \tag{E1}$$
$$\gamma_{eff} = \frac{4 \times k_{particle} \times D \times \rho_0 \times N_A}{6 \times \bar{c} \times M} \tag{E2}$$
$$\Delta\gamma_{eff} = \gamma_{eff,2FT} - \gamma_{eff,1FT} \tag{E3}$$
**3 Results and discussion**
**3.1 Heterogeneous reactions rates of organic aerosols upon ethylamine exposure**
The $O_3$ addition reactions to the C=C bonds of Sqe in the first flowtube reactor generate primary ozonides (POZs) (Heine et
al., 2017), as illustrated in Fig. S8. These POZs subsequently decompose to form three ketones ($C_3H_6O$, $C_8H_{14}O$, and $C_{13}H_{22}O$)
with molecular weights ($MWs$) of 58, 126, and 194, three aldehydes ($C_{17}H_{28}O$, $C_{22}H_{36}O$, and $C_{27}H_{44}O$) with $MWs$ of 248, 316,





and 384, and six Criegee intermediates (CIs) (Heine et al., 2017). Unimolecular isomerization reactions of $C_{17}$, $C_{22}$, and $C_{27}$
CIs produce carboxylic acids ($C_{17}H_{28}O_2$, $C_{22}H_{36}O_2$, and $C_{27}H_{44}O_2$) with *MWs* of 264, 332, and 400 (Arata et al., 2019; Zahardis
et al., 2005). Bimolecular reactions between CIs and aldehydes (or ketones) generate a series of SOZs (Fig. S9), including $C_6$,
$C_{11}$, $C_{16}$, $C_{20}$, $C_{21}$, $C_{25}$, $C_{26}$, $C_{30}$, $C_{34}$, $C_{35}$, $C_{39}$, $C_{40}$, $C_{44}$, $C_{49}$, and $C_{54}$ SOZs (Heine et al., 2017). Considering the relative higher
partitioning of smaller species (e.g., ketones and smaller SOZs) into the gas phase, the present work mostly focused on larger
SOZs, aldehydes, and carboxylic acids.
Figure 2a illustrates the relative abundance of $C_{20}$ to $C_{54}$ SOZs as a function of ethylamine exposure. These SOZs exhibit
distinct heterogeneous decay rates ($k_{particle}$), with the $C_{21}$ SOZ exhibiting the highest rate. The differential effective uptake
coefficients ($\Delta\gamma_{eff}$), quantifying the contribution of SOZ reactions with ethylamine in the secondary flowtube reactor, were
then calculated using Equations E2 and E3 (Fig. 2c). The $\Delta\gamma_{eff}$ values of SOZs generally show consistent tendencies with the
decay kinetics and exhibit a zigzag pattern that decreases with increasing carbon chain length of SOZs.

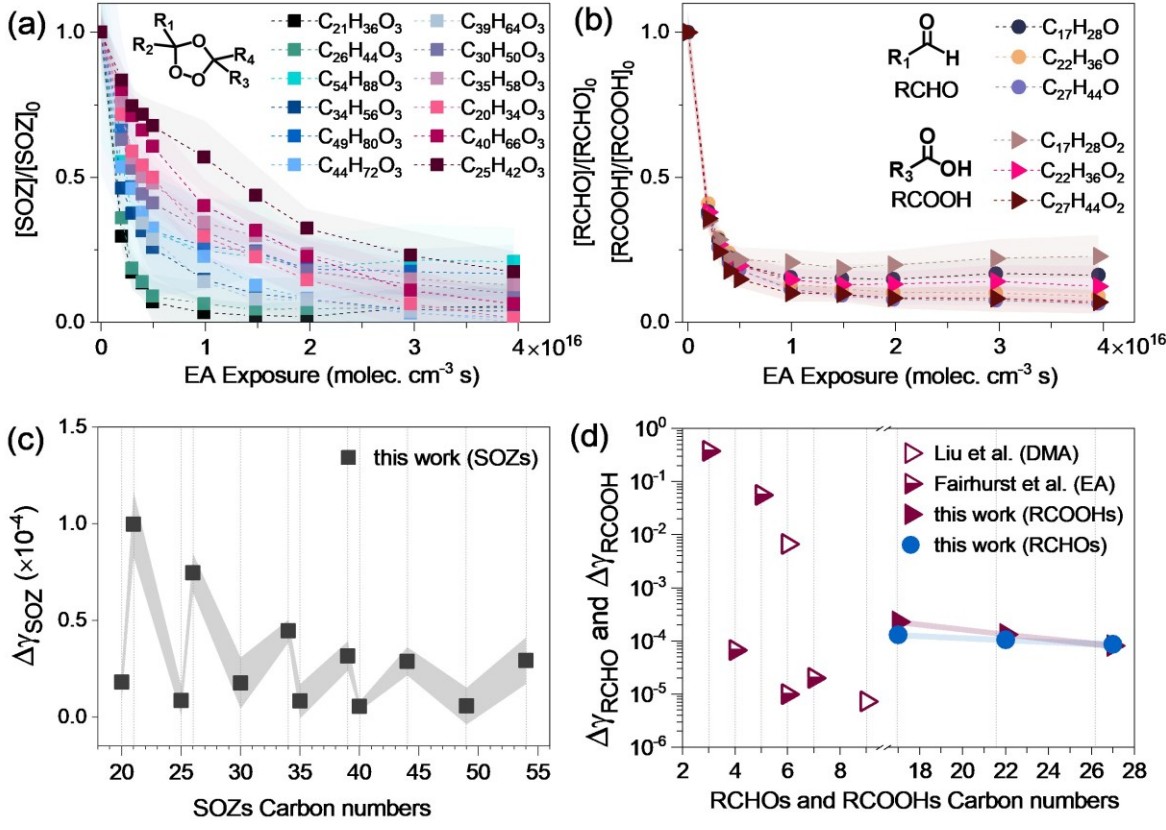


**Figure 2: (a-b) Decay of SOZs, aldehydes, and carboxylic acids as a function of ethylamine (EA) exposure in tandem flowtube experiments. (c-d) Differential effective uptake coefficients ($\Delta\gamma$) for $C_{20}$, $C_{21}$, $C_{25}$, $C_{26}$, $C_{30}$, $C_{34}$, $C_{35}$, $C_{39}$, $C_{40}$, $C_{44}$, $C_{49}$, and $C_{54}$ SOZs; $C_{17}$, $C_{22}$, and $C_{27}$ carboxylic acids, and $C_{17}$, $C_{22}$, and $C_{27}$ aldehydes. Uptake coefficients for carboxylic acids from Refs.(Liu et al., 2012; Fairhurst et al., 2017a) are included.**




Differences in the initial concentrations of SOZs may contribute to their distinct heterogeneous reactivities ($\Delta\gamma$) shown in Fig.
2c. As demonstrated by Heine et al. (2017), in multi-component particles, the heterogenous reactivity of each component
depends on its initial concentration (Jacobs et al., 2016; Zeng et al., 2020; Arata et al., 2019). Heine et al. (2017) reported that
the abundance of SOZs formed during the ozonolysis of squalene varies, following the order: $C_{30} > C_{25}$, $C_{35} > C_{44} > C_{20}$, $C_{21}$,
$C_{39}$, $C_{40}$, $C_{49} > C_{26}$, $C_{34}$, $C_{54}$ SOZs, as illustrated in Fig. S7a. Consequently, in this work, the initial concentrations of SOZs
formed from the ozonolysis of Sqe in the flowtube reactor differ. To quantify the influence of the initial SOZ concentrations
on their heterogeneous reaction rates, the $\Delta\gamma_{eff}$ values for SOZs were normalized for their corresponding initial concentrations.
As illustrated in Fig. S7b, the normalized $\Delta\gamma$ values exhibit smaller differences compared to the original $\Delta\gamma_{eff}$ values. This
observation supports the hypothesis that differences in initial SOZ concentrations affect their decay rate upon ethylamine
exposure. Additionally, SOZs with long-chain substituents (e.g., $C_{54}$ SOZ) exhibit lower reactivity (Ponec et al., 1997). This
reduced reactivity may be attributed to the steric hindrance effects (Hon et al., 1995), which restrict the conformational
flexibility of SOZ molecules during attack by ethylamine.
Figure 2b illustrates the decay kinetics of representative aldehydes and carboxylic acids. These aldehydes and carboxylic acids
decay faster than SOZs. Consistently, their differential effective uptake coefficients ($\Delta\gamma_{eff}$ from $10^{-5}$ to $10^{-4}$) are larger than
those of SOZs ($10^{-5}$ to $10^{-6}$), as illustrated in Fig. 2d. This difference could be explained by the higher acidity of carboxylic
acids, which enhances the heterogeneous reactions between these acids upon ethylamine exposure. As reported by Liu et al.
(2012) the heterogeneous uptake coefficients of dimethylamine ($C_2H_7N$, an isomer of ethylamine), with citric acid (a triacid,
$C_6H_8O_7$, $\gamma \sim 10^{-3}$) is significantly larger than with humic acid (a diacid, $C_9H_9NO_6$, $\gamma \sim 10^{-6}$). They attributed this difference to
the stronger acidity of citric acid relative to humic acid.
Figure 2d illustrates that the measured $\Delta\gamma_{eff}$ values for carboxylic acids follow the trend: $C_{17}H_{28}O_2 > C_{22}H_{36}O_2 > C_{27}H_{44}O_2$,
indicating a negative dependence of heterogeneous reactivity on carbon chain length. A similar trend is observed for aldehydes,
i.e., $C_{17}H_{28}O > C_{22}H_{36}O > C_{27}H_{44}O$. To our knowledge, no prior experimental measurements exist for the heterogeneous
reactivity of aldehyde particles with ethylamine, whereas reactivity trends for carboxylic acids of varying carbon chain length
have been investigated (Fairhurst et al., 2017a). For example, Fairhurst et al. (2017a) measured the heterogeneous reactivities
of ethylamine by solid dicarboxylic acids with varying carbon numbers: malonic acid ($C_3H_4O_4$), succinic acid ($C_4H_6O_4$),
glutaric acid ($C_5H_8O_4$), adipic acid ($C_6H_{10}O_4$), and pimelic acid ($C_7H_{12}O_4$). Their measured uptake coefficients are
approximately $10^{-1}$ for $C_3$ diacids, $7 \times 10^{-5}$ for $C_4$ diacid, and $1 \times 10^{-5}$ for the $C_6$ diacid. They attributed this trend to differences
in the crystalline surface structures of these solid acids. Thus, these reported uptake coefficients for carboxylic acids reacting
with amine span a wide range ($10^{-1}$ to $10^{-6}$), suggesting that more comprehensive data are needed to elucidate the reactivity
trends for both carboxylic acids and aldehydes across varying carbon chain length.



## 3.2 Products distribution during the heterogeneous reactions of organic particles

Figure 3a illustrates the product distribution during the ozonolysis of Sqe aerosols. Consistent with prior observations (Liu et al., 2024), maximum product yields were achieved at approximately 60% Sqe conversion (corresponding to 0.678 ppm $O_3$), with representative products (SOZs, aldehydes, and acids) shown in Figs. S2 and S3. Figures 3b and S3 shows representative products from heterogeneous reactions between SOZs with ethylamine ($C_2H_7N$), with mass spectral analysis revealing three product classes. First, protonated molecular ions (denoted as $[M+H]^+$) appear at *m/z* 314, 368, 382, 436, 450, 504, 558, 572, 626, 640, 694, and 762, corresponding to adducts from reactions between SOZs and ethylamine, i.e., SOZs + $C_2H_7N$. For instance, the *m/z* 504 peak corresponds to $C_{32}H_{57}O_3N$ (*MW* 503) from the reaction of $C_{30}$ SOZ ($C_{30}H_{50}O_3$) with $C_2H_7N$. Detailed reaction mechanisms are discussed in Sect. 3.3. Second, the deprotonated ions ($[M-H]^-$) at *m/z* 294, 348, 362, 416, 430, and 484 derive from subsequent dehydration products from SOZs + $C_2H_7N$ reactions, i.e., SOZs + $C_2H_7N$ – $H_2O$. For instance, the *m/z* 484 peak matches the $H_2O$ loss product of $C_{32}H_{57}O_3N$, i.e., $C_{32}H_{57}O_3N$ – $H_2O$. Third, the deprotonated ions ($[M-H]^-$) at *m/z* 278, 332, 346, 400, 414, 468, and 536 correspond to the $H_2O_2$ elimination products from SOZs and $C_2H_7N$ reactions, i.e., SOZs + $C_2H_7N$ – $H_2O_2$. For instance, the *m/z* 468 peak represents the $H_2O_2$ elimination product of $C_{32}H_{57}O_3N$, i.e., $C_{32}H_{57}O_3N$ – $H_2O_2$. Notably, dehydration and $H_2O_2$ elimination products were only observed for smaller SOZs ($C_n$ < 35), suggesting that substituent effects and chain length of SOZs could influence the amination reaction pathways (see Sect. 3.2 for mechanistic details).



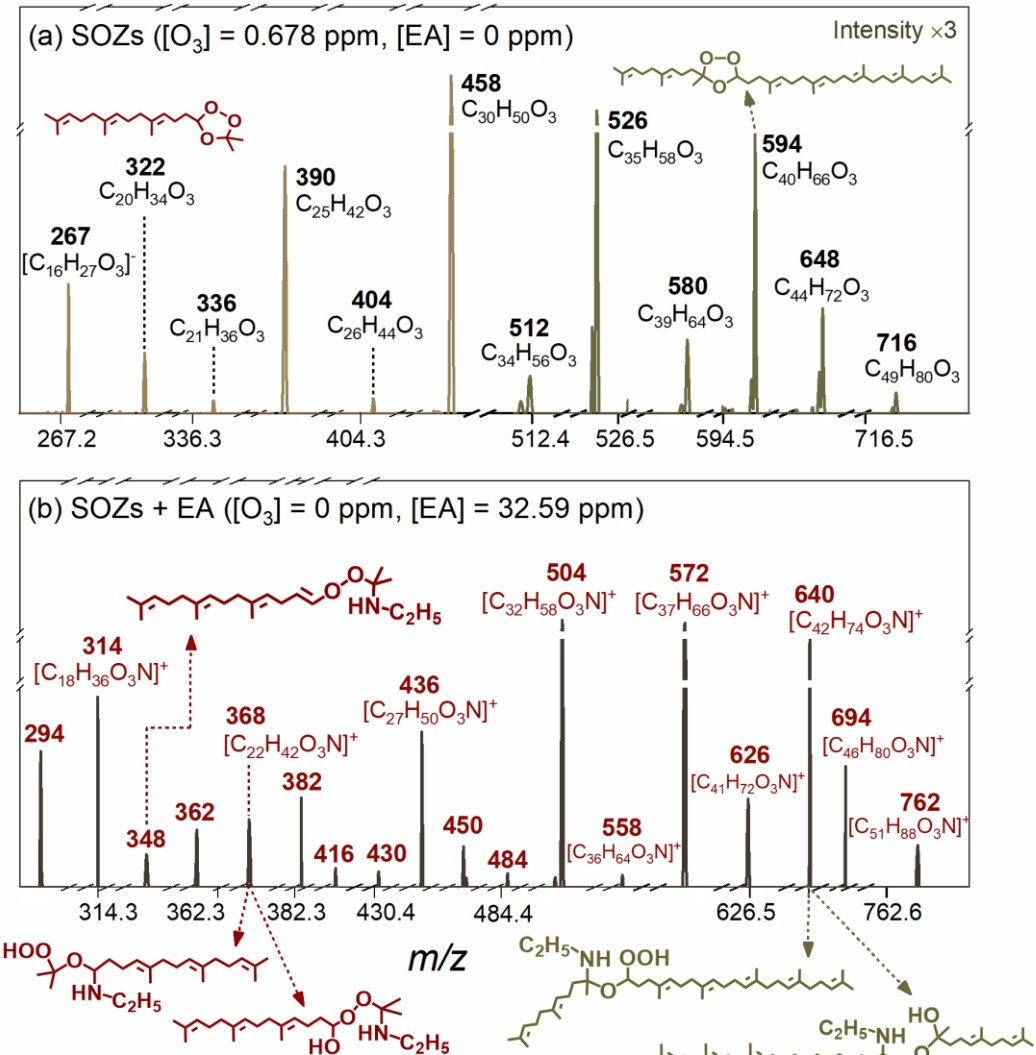

**Figure 3: Mass spectra of (a) representative SOZs formed during the ozonolysis of Sqe in the first flowtube reactor, and (b) their amination products upon exposure to ethylamine (EA) in the secondary flowtube reactor.**

Figures S2b and S3b illustrates the products formed from heterogeneous reactions of $C_{17}$, $C_{22}$, and $C_{27}$ aldehydes (RCHOs), as well as $C_{17}$, $C_{22}$, and $C_{27}$ carboxylic acids (RCOOHs) upon ethylamine exposure in the secondary flowtube reactor. The protonated molecular ions ([M+H]$^+$) at $m/z$ 294, 310, 362, 378, 430, and 446 correspond to products from reactions of these aldehydes (or carboxylic acids) with ethylamine ($C_2H_7N$), i.e., RCHOs (or RCOOHs) + $C_2H_7N$ (Bain et al., 2016; De Haan et al., 2011). For instance, the peak at $m/z$ 294 is assigned to $C_{19}H_{35}ON$ (*MW* 293) formed from the reaction of $C_{17}$ aldehyde ($C_{17}H_{28}O$) with $C_2H_7N$. These RCHOs (or RCOOHs) + $C_2H_7N$ adducts subsequently undergo $H_2O$ elimination reactions (Shen et al., 2023; Bain et al., 2016; De Haan et al., 2009; Tuguldurova et al., 2024), producing characteristic peaks at $m/z$ 276, 344,





and 412 (from aldehyde reactions); as well as *m/z* 292, 360, and 428 (from acid reactions). For instance, the $[M+H]^+$ peak at
*m/z* 276 corresponds $C_{19}H_{33}N$ (*MW* 275), resulting from $H_2O$ elimination from $C_{19}H_{35}ON$, i.e., $C_{19}H_{35}ON - H_2O$.

### 3.3 Amination mechanisms of secondary ozonides, carboxylic acids, and aldehydes

A reaction mechanism was developed to elucidate the heterogeneous reactions of organic aerosols, including SOZs, carboxylic
acids ($C_{17}H_{28}O_2$, $C_{22}H_{36}O_2$, and $C_{27}H_{44}O_2$), and aldehydes ($C_{17}H_{28}O$, $C_{22}H_{36}O$, and $C_{27}H_{44}O$).
For SOZs, the electronegativity of neighboring oxygen atoms induces a net positive charge on the α-carbon atoms (Fig. 4),
facilitating nucleophilic attack by an amine (Jørgensen and Gross, 2009). This could lead to the formation of either hydroxyl
peroxyamines (R1) or amino hydroperoxides (R2), depending on the nucleophilic attack sites (Zahardis et al., 2008; Jørgensen
and Gross, 2009; Almatarneh et al., 2020). However, these competing pathways and their proposed products remain
controversial (Fig. S11). Zahardis et al. (2008) proposed that SOZ reaction with octadecylamine generates a hydroxyl
peroxyamine intermediate, which subsequently decomposes to nonanal and a $C_{27}$ amide via $H_2O$ elimination (Fig. S11a).
Conversely, Almatarneh et al. (2020), Jørgensen and Gross (2009), and Na et al. (2006) suggested that SOZ reactions with
ammonia form amino hydroperoxide intermediates (Figs. S11b to S11e). Notably, neither hydroxyl peroxyamine nor amino
hydroperoxide intermediates have been experimentally detected in these studies (Zahardis et al., 2008; Almatarneh et al., 2020;
Jørgensen and Gross, 2009; Na et al., 2006). Contrasting both pathways, Qiu et al. (2024) demonstrated that ethylamine
addition to a cyclic SOZ directly yields a linear amination product through simultaneously $H_2O$ elimination (Fig. S11a), i.e.,
bypassing formation of either proposed intermediates (Fig. S11f).



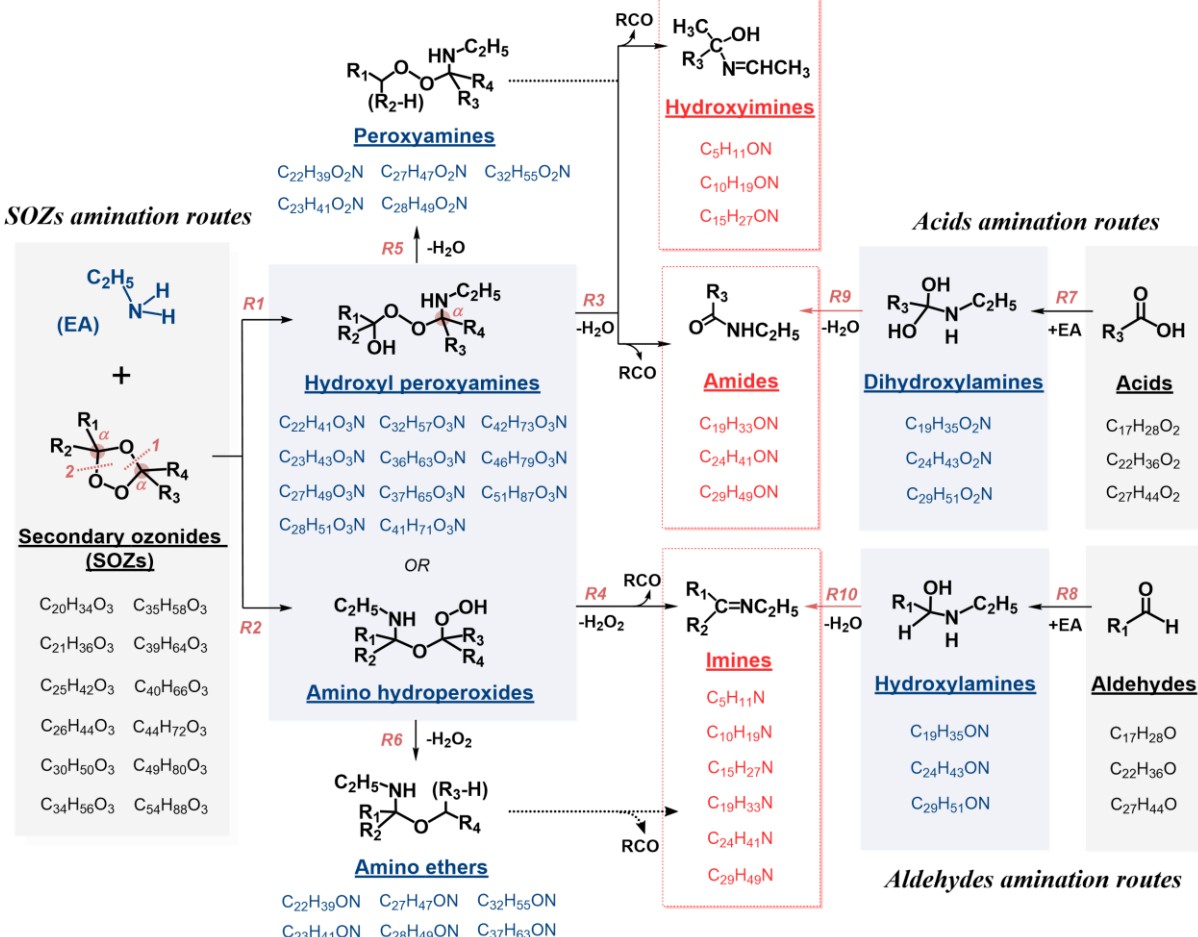

**Figure 4: Amination mechanisms of secondary ozonides (SOZs), carboxylic acids (RCOOHs), and aldehydes (RCHOs) upon ethylamine (EA) exposure, with representative chemical structures and compositions of reactants and products.**

In this work, mass peaks corresponding to SOZs + ethylamine ($C_2H_7N$) adducts were observed (Fig. 3b), providing direct experimental evidence for nucleophilic attack by ethylamine on SOZs. The MS$^2$ spectra provide complementary evidence for their structural characterization. Figure 5 illustrates representative MS$^2$ fragmentation patterns of the $[C_{32}H_{58}O_3N]^+$ ion (denoted as $[M+H]^+$), corresponding to the $C_{32}H_{57}O_3N$ product formed from $C_{30}$ SOZ ($C_{30}H_{50}O_3$) + ethylamine ($C_2H_7N$) reaction. Two representative isomers, a hydroxyl peroxyamine (I) and amino hydroperoxide (II), were selected for analysis (Fig. 5a). Both isomers undergo $C_2H_5NH$ elimination, yielding $[M-C_2H_6N]^-$ ions ($m/z$ 459). The hydroxyl peroxyamine isomer (I) subsequently loses $C_3H_7O$ and $C_{11}H_{18}$ to form $m/z$ 294, while the amino hydroperoxide isomer (II) eliminates $HO_2$ yielding $m/z$ 426. Additional fragmentation peaks ($m/z$ 60, 86, 149, 383, and 441) may derive from these isomers. It is also noted that $C_{30}$ SOZ isomers formation during the ozonolysis of Sqe (Figs. S9 and S10b) suggests additional $C_{32}H_{57}O_3N$ isomers likely contribute to these fragmentation patterns (Fig. 5). Figure 6a illustrates the kinetics of representative $C_{30}$ hydroxyl peroxyamines (or amino hydroperoxides) as a function of ethylamine exposure.




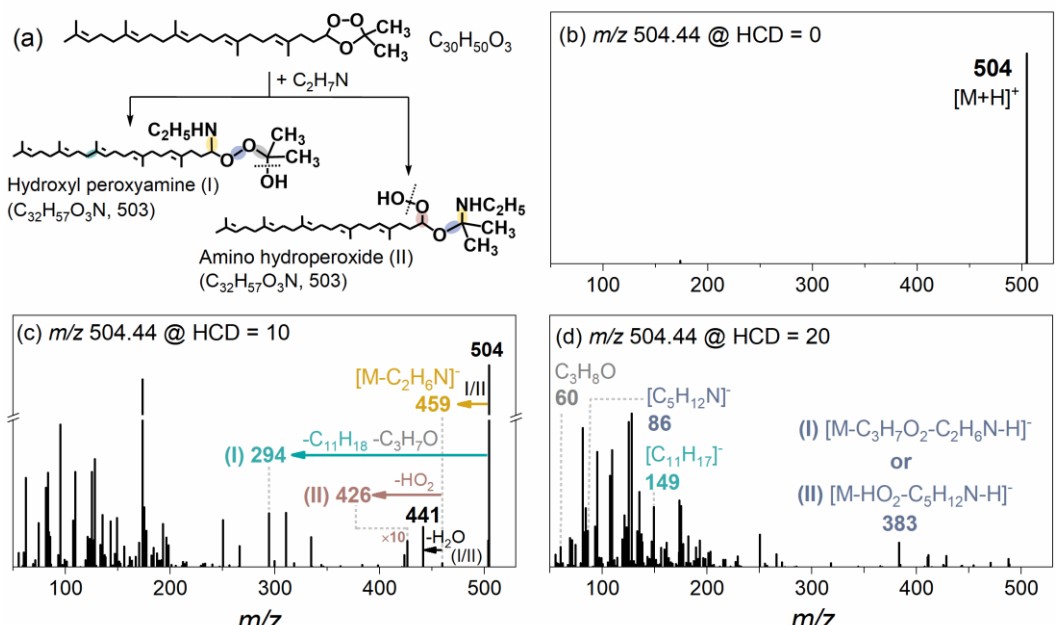

**Figure 5: (a) Structures of two representative isomers ($C_{32}H_{57}O_3N$, *MW* 503), *i.e.,* hydroxyl peroxyamine (I) and amino hydroperoxide (II). (b-d) MS$^2$ fragmentation of their protonated ions at HCD energies: (b) 0, (c) 10%, and (d) 20 %.**

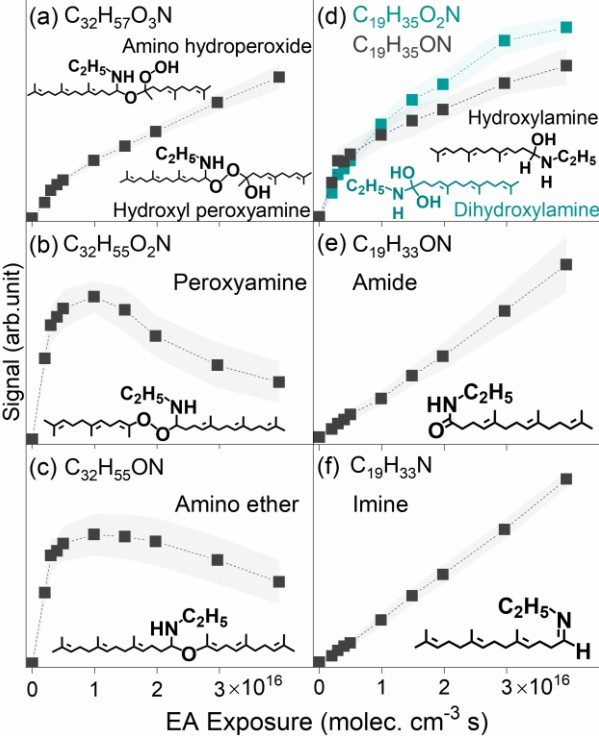

**Figure 6: Relative abundances as a function of ethylamine (EA) exposure for: (a) hydroxyl peroxyamine (or amino hydroperoxide), (b) peroxyamine, (c) amino ether, (d) dihydroxylamine and hydroxylamine, (e) amide, and (f) imine.**



Consumption reactions for the amino hydroperoxide and hydroxyl peroxyamine intermediates were previously proposed
(Zahardis et al., 2008; Na et al., 2006; Almatarneh et al., 2020; Jørgensen and Gross, 2009). Amino hydroperoxide
decomposition has be demonstrated to generate smaller imines and aldehydes via $H_2O_2$ elimination (Fig. 4). For instance,
Almatarneh et al. (2020) demonstrated that a $C_2$ amino hydroperoxide decomposition yields methylenimine, formaldehyde,
and $H_2O_2$ (Fig. S11). Consistently, products with compositions of $C_5H_{11}N$, $C_{10}H_{19}N$, $C_{15}H_{27}N$, $C_{19}H_{33}N$, $C_{24}H_{41}N$, and $C_{29}H_{49}N$
are assigned to be imines (Fig. S13) that could be attributed to these reactions. In contrast, for hydroxyl peroxyamine, Zahardis
et al. (2008) reported that hydroxyl peroxyamine decomposition generates a $C_{27}$ amide, $C_9$ aldehyde with $H_2O$ elimination.
Observed products with compositions of $C_{19}H_{33}ON$, $C_{24}H_{41}ON$, and $C_{29}H_{49}ON$ are proposed to be amides (Fig. S12) that could
be formed from these pathways. Moreover, hydroxyl peroxyamines lacking α-H atoms form hydroxyimines ($C_5H_{11}ON$,
$C_{10}H_{19}ON$, and $C_{15}H_{27}ON$), rather than amides. For instance, a $C_{36}$ hydroxyl peroxyamine with available α-H atom (Fig. S12a)
decomposes to a $C_{19}$ amide, $C_{17}$ aldehyde and $H_2O$, whereas a $C_{28}$ hydroxyl peroxyamine without α-H atom (Fig. S12b) yields
$C_{15}$ hydroxyimine, $C_{13}$ ketone, and $H_2O$.
These hydroxyimines, amides, and imines (Fig. 4) exhibit significantly lower carbon numbers than their precursor hydroxy
peroxyamines (R3) or amino hydroperoxides (R4). Consequently, neither R3 nor R4 pathways explain observed products
retaining same carbon numbers with amino hydroperoxides or hydroxy peroxyamines. This work therefore proposes two new
pathways: R5 (hydroxyl peroxyamine → $H_2O$ + peroxyamine), and R6 (amino hydroperoxide → $H_2O_2$ + amino ether). With
$C_{32}$ intermediates for example, these pathways reasonably explain observed $C_{32}O_{55}O_2N$ (peroxyamine) and $C_{32}H_{55}ON$ (amino
ether) products formed from $C_{32}O_{57}O_3N$ via $H_2O$ or $H_2O_2$ elimination, respectively (Figs. 6b and 6c). The observed trend of
their relative abundance, which initially increased and then decreased as a function of ethylamine exposure, providing evidence
for mediating the conversion of hydroxy peroxyamine (or amino hydroperoxide) to hydroxylamine, amide, and imine. Given
the absence of these pathways in prior work (Almatarneh et al., 2020; Na et al., 2006; Zahardis et al., 2008; Qiu et al., 2024;
Jørgensen and Gross, 2009), these R5 and R6 requires further investigations.
Figure 4 illustrates reaction mechanisms for ethylamine reactions with $C_{17}$, $C_{22}$, and $C_{27}$ carboxylic acids, as well as $C_{17}$, $C_{22}$,
and $C_{27}$ aldehydes, forming dihydroxylamines (R7) and hydroxylamines (R8), respectively (De Haan et al., 2011; Ditto et al.,
2022; Bain et al., 2016; Shashikala et al., 2023; Sarkar et al., 2019). For instance, the nucleophilic attack reaction of ammonia
at carbonyl (C=O) site of acetaldehyde generates a hydroxylamine as demonstrated by Sarkar et al. (2019). Here,
dihydroxylamines ($C_{19}H_{35}O_2N$, $C_{24}H_{43}O_2N$, $C_{29}H_{51}O_2N$) and hydroxylamines ($C_{19}H_{35}ON$, $C_{24}H_{43}ON$, and $C_{29}H_{51}ON$) have
been measured by APPI-HRMS (Fig. S2b) (Sarkar et al., 2019), as well as their kinetics as a function of ethylamine exposure
(e.g., $C_{19}$ dihydroxylamine and hydroxylamine in Fig. 6d). Subsequent $H_2O$ elimination reactions of dihydroxylamines and
hydroxylamines yield amides (R9, $C_{19}H_{33}ON$, $C_{24}H_{41}ON$, and $C_{29}H_{49}ON$) and imines (R10, e.g., $C_{19}H_{33}N$, $C_{24}H_{41}N$, and
$C_{29}H_{49}N$) (Montgomery and Day, 1965), respectively, as shown in Fig. S2b. For example, $C_{17}$ aldehyde ($C_{17}H_{28}O$, Fig. S14a)
reacts with ethylamine to generate $C_{19}$ hydroxylamine intermediate ($C_{19}H_{35}ON$, *MW* 293), which dehydrates to $C_{19}$ imine





($C_{19}H_{33}N$, *MW* 275). Additionally, these amides and imines can be also produced from hydroxyl peroxyamines (R3) or amino
hydroperoxides (R4), as shown in Figs. 6e and 6f.

## 4 Conclusions

Atmospheric amines critically modulate the evolution of aerosols and particulate pollution. Here, we investigate heterogeneous
reactions of ethylamine with SOZs, carboxylic acids, and aldehydes aerosols using a tandem flowtube reactor combined with
online APPI-HRMS. Heterogeneous reactivities ($\Delta\gamma_{eff}$) for SOZs decrease 17.5-fold with increasing carbon chain length, from
$\Delta\gamma_{eff} = 1.0 \times 10^{-4}$ for $C_{21}$ SOZ to $5.7 \times 10^{-6}$ for $C_{49}$ SOZ, with nonmonotonic behavior suggesting substitution effects. Crucially,
reactions of ethylamine with carboxylic acids and aldehydes exhibit $\Delta\gamma_{eff}$ values of $10^{-4}$ to $10^{-5}$, exceeding SOZs reactivity at
equivalent carbon numbers, with reactivity similarly declining with chain length. This reactivity dependence implies
atmospheric lifetimes ($\tau \propto \gamma^{-1}$) spanning two orders of magnitude across these organic aerosols, thereby controlling their
differential impacts on air quality, health, and climate.
Moreover, hydroxyl peroxyamines, amino hydroperoxides, peroxyamines, and amino ethers, were measured as characteristic
intermediates linking SOZ consumption to stable nitrogenous products (hydroxyimines, amides, and imines).
Dihydroxylamines and hydroxylamines from reactions of carboxylic acids and aldehydes were characterized as crucial
intermediates. Further reaction mechanism analysis reveals that nucleophilic addition of ethylamine to SOZs initiates the
formation of hydroxyl peroxyamines and amino hydroperoxides. Beyond established cleavage pathways yielding form
hydroxyimines, amides, and imines, this work demonstrates new elimination pathways: hydroxyl peroxyamines (amino
hydroperoxides) $\rightarrow H_2O$ (or $H_2O_2$) + peroxyamines (or amino ethers). These mechanistic insights elucidate the transformation
of organic aerosols to nitrogen-containing secondary organic aerosols (SOAs), providing fundamental parameters for more
accurate modeling of atmospheric processes.

**Data availability.** The authors confirm that the data supporting the findings of this work are available within the article and
its supplementary information.

**Supplement.** Supplementary details about the experimental data (Figs. S1 to S7) and reaction mechanism (Figs. S8 to S14).

**Author contributions.** Peiqi Liu: Investigation, Data curation, Formal analysis, Writing - original draft. Jigang Gao and
Yulong Hu: Data curation. Wenhao Yuan, Zhongyue Zhou and Fei Qi: Methodology. Meirong Zeng: Conceptualization,
Methodology, Supervision, Writing - review & editing.

**Competing interests.** The authors declare no competing financial interest.





**Acknowledgments.** The authors are grateful for the funding support from the National Natural Science Foundation of China and Shanghai Science and Technology Innovation Action Plan. The authors thank Kevin R. Wilson (Lawrence Berkeley National Laboratory) for helpful discussions.

**Financial support.** This study was supported by the National Natural Science Foundation of China (22373066, 52376119) and Shanghai Science and Technology Innovation Action Plan (24142201500).

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





**Graphic abstracts**

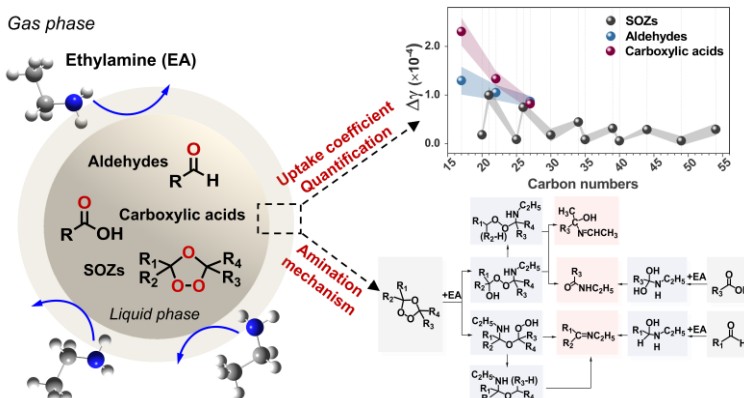

