# Peer review of "Ethylamine-Driven Amination of Organic Particles: Mechanistic 2 Insights via Key Intermediates Identification"

_EGUsphere, 2025_

## Author Comment (AC1)

**RC1 Comments:**

The manuscript is well-written, and conclusions are well-supported by the experimental data. I recommend this work for publication, and some minor comments can be considered.

**Q1-1**. In the introduction, the authors justify the use of squalene as a model system. However, a brief statement for selecting ethylamine as the model amine, compared to more commonly studied amines (e.g., ammonia or dimethylamine), can be helpful. Is it due to its atmospheric relevance, reactivity, or as a simpler model for primary amines?

**Reply1-1**: Ethylamine is abundant in the atmosphere (Atmos. Environ. 2013, 71, 95-103). It exhibits higher heterogeneous reactivity towards the aerosols studied in this work. Its simple structure also makes it a good model for studying primary amination mechanisms, avoiding the complexity of secondary/tertiary amine groups. We have added a justification at Line 67 on Page 3 of the revised manuscript as follows.

"As a representative atmospheric amine (Lee and Wexler, 2013), ethylamine was selected for its remarkable heterogeneous reactivity and simple structure."

**Q1-2**. The use of APPI-HRMS is critical to the identification of intermediates and products. To facilitate a better assessment of the methodology, it is suggested to provide key instrumental parameters in the supplementary information. These should include, but not be limited to: the ionization mode used (positive/negative), mass resolution, mass range scanned, and vaporizer temperature.

**Reply1-2**: We have added more details about the APPI-HRMS on Page S5 in the revised *Supplementary Information* as follows.

"The mass resolution of HRMS (Orbitrap Fusion, Thermo Scientific) is 500,000 at m/z 200. During the experiments, the MS mode was set to positive polarity. Meanwhile, the ion transfer tube was maintained at 300 °C. The mass range from m/z 75 to 800 was scanned with a rate of 30 spectra/min, and the resulting data were collected using Xcalibur 4.0 software."

**Q1-3**. The mass spectrum in Figure 3b would be more clearly distinguishable if the different product classes (adducts,  $-H_2O_2$ ) were labeled with different colors or symbols directly on the spectrum.

**Reply1-3**: Figure 3b along with its corresponding caption has been revised in the manuscript as follows.

Figure 3: Mass spectra of (a) representative SOZs formed during the ozonolysis of Sqe in the first flowtube reactor, and (b) their amination products (adducts of SOZ + EA are in red and dehydration products are in blue) upon exposure to ethylamine (EA) in the secondary flowtube reactor.

**Q1-4**. The identification of the four key intermediates is a core finding of this work. For clarity, it is suggested to add a simplified schematic flowchart, either in Figure 4 or as a supplementary figure, to summarize the competing pathways and subsequent consumption pathways. This would facilitate visualization of the complex reaction network.

**Reply1-4**: A simplified schematic flowchart has been added as Fig. S15 in the revised *Supplementary Information*. Meanwhile, an introduction for this Fig. S15 has been added at Line 282 on Page 15 in the revised manuscript.

"Figure S15 summarizes a simplified mechanism involving four key intermediates, including hydroxyl peroxyamines, peroxyamines, amino hydroperoxides, and amino ethers."

Figure S15. A simplified mechanism of SOZs upon EA exposure.

**Q1-5**. For the newly proposed intermediates (peroxyamines and amino ethers), it is suggested to provide their exact mass and compare them with the experimentally observed m/z values in a supplementary table. Listing representative intermediates and products observed for each SOZ in this table can provide additional evidence for their identification.

**Reply1-5**: A new Table S1 has been added to the revised *Supplementary Information*, which lists the representative products, including their molecular formulas, representative structures, theoretical exact mass, and experimentally observed m/z.

**Technical corrections:**

There are inconsistencies in the terminology used for differential effective uptake coefficients, which are sometimes written as " $\Delta \gamma_{eff}$ ", and sometimes abbreviated as " $\Delta \gamma$ ".

**Reply1-6:** To ensure consistency, we have represented the differential effective uptake coefficients as  $\Delta \gamma_{\text{eff}}$  in the revised manuscript and *Supplementary Information*.

Line 203, "see Sect. 3.2 for mechanistic details," but the mechanistic details are in Sect. 3.3. Please correct it.

**Reply1-7:** This word "Sect. 3.2" has been revised to "Sect. 3.3" on Page 9 of the revised manuscript.

Line 254, "has be demonstrated", this should be correct to "has been demonstrated".

**Reply1-8:** We have corrected this.

Line 268, " $C_{32}O_{55}O_2N$  (peroxyamine)" should be corrected to  $C_{32}H_{55}O_2N$ .

**Reply1-9:** We have corrected this to " $C_{32}H_{55}O_2N$ ".

Line 269, " $C_{32}O_{57}O_3N$ " should be corrected to " $C_{32}H_{57}O_3N$ ".

**Reply1-10:** We have corrected this to " $C_{32}H_{57}O_3N$ ".

---

## Author Comment (AC2)

**RC2 comments:**

I recommend publication of this work with minor revisions.

- **Q2-1:** It is suggested to provide the material of the tandem flow-tube (e.g., Pyrex, stainless steel).
  - **Reply2-1**: The material information (quartz flowtube reactors) has been added at Line 90 and Line 107 of the revised manuscript as follows.
  - "...the first quartz flowtube reactor ... a secondary quartz flowtube reactor"
- **Q2-2:** The range of  $O_3$  concentration is not mentioned in the paper. It is suggested that this parameter be clarified, as it is critical for the SOZ formation.
  - **Reply2-2**: The range of O3 concentration has been added at Line 95 on Page 4 of the revised manuscript as follows.
  - "The O3 concentration in the first flowtube reactor was varied from 0 to 0.678 ppm."
- **Q2-3:** A supplementary table, listing values of  $\gamma_{eff,2FT}$  and  $\gamma_{eff,1FT}$  for representative SOZs, carboxylic acids, and aldehydes, can be useful to add.
  - **Reply2-3**: A new supplementary Table S2, listing values of  $\gamma_{eff,2FT}$ ,  $\gamma_{eff,1FT}$  and  $\Delta\gamma_{eff}$  for representative SOZs, carboxylic acids, and aldehydes has been added to the *Supplementary Information*. Meanwhile, an introduction for this Table S2 has been added at Line 158 on Page 6 of the revised manuscript as follows.
  - "...calculated using Equations E2 and E3 (Fig. 2c and Table S2)."
- **Q2-4:** The curves for different intermediates (e.g., dihydroxylamine vs. hydroxylamine) in Figure 6 are distinguished only by color. Adding distinct symbol types would improve the clarity of the figure.
  - **Reply2-4**: In the revised manuscript, Figure 6d has been updated by using different symbol types for dihydroxylamine and hydroxylamine, as follows.

Figure 6: Experimental signal as a function of ethylamine (EA) exposure for: (a) hydroxyl peroxyamine (or amino hydroperoxide), (b) peroxyamine, (c) amino ether, (d) dihydroxylamine and hydroxylamine, (e) amide, and (f) imine.

**Q2-5:** It seems that the caption for Figure 6 mentions "Relative abundance", which does not align with the y-axis label "Signal (arb. unit)". Please update the caption to ensure the consistency between the figure and its description.

**Reply2-5**: The phrase "Relative abundance" in the caption of Fig. 6 has been revised to "Experimental signal" on Page 15 of the revised manuscript.

**Q2-6:** Line 30, the phrase "initiates through the nucleophilic attack" should be grammatically expressed as "is initiated by a nucleophilic attack".

**Reply2-6:** The phrase "initiates through the nucleophilic attack" has been revised to "is initiated by a nucleophilic attack" in Line 32 of the revised manuscript.

**Q2-7:** Line 38, "ethylamine and on a  $C_{15}$  SOZ" contains a superfluous conjunction. For clarity, this should be corrected to "ethylamine on a  $C_{15}$  SOZ".

**Reply2-7:** The phrase "ethylamine and on a  $C_{15}$  SOZ" has been revised to "ethylamine on a  $C_{15}$  SOZ" in Line 40 of the revised manuscript.

**Q2-8:** Line 116-117, the sentence "the net contribution of heterogeneous reactions... were quantitatively determined" has a subject-verb agreement error. Please correct it to "the net contribution... was quantitatively determined."

Reply2-8: The verb "were" has been corrected to "was" in Line 122 of the revised

manuscript.

**Q2-9:** Check the capitalization style of the article titles in the references. It seems that there is an inconsistency in the use of capitalization, specifically regarding sentence case *vs.* title case. It is recommended to standardize the format according to the journal's guidelines.

**Reply2-9:** The capitalization of references titles in the revised manuscript and *Supplementary Information* has been standardized according to the *Atmospheric Chemistry and Physics* style.

**Q2-10:** Both relative humidity and pH are critical factors influencing the heterogeneous reactions involving amines. Thus, if the reaction system has recorded humidity and pH data, it is advisable to explicitly state them in the manuscript.

**Reply2-10:** The experiments were conducted under dry conditions. According to Heine et al. (Environ. Sci. Technol. 2017, 51, 13740), the relative humidity (RH) under similar dry conditions correspond to approximately 3%. As for pH value, it was not recorded. This RH information has been added at Line 113 on Page 5 of the revised manuscript as follows.

"The experiments have been conducted under dry condition, corresponding to a relative humidity of approximately 3% (Heine et al., 2017)."

---

## Author Response (AR2)

**Comments from the Editor:**

***Q1****. ACP does not require a Graphic abstract. You can remove it from your manuscript.*

    **Reply1**: The graphic abstract has been removed from the manuscript.

***Q2****. In Figure 3, there is a note "intensity\*3". It is not clear if it is only for m/z 594 or for a region. It does not make sense if it is only for ion 594.*

    **Reply2**: In Figure 3a, the note "intensity × 3" indicates signal amplification for the *m/z* range of 500-800, as illustrated in the revised Figure 3 and its caption in the revised manuscript.

[Figure]

Figure 3: Mass spectra of (a) representative SOZs (the signals within the *m/z* range of 500-800 are amplified) formed during the ozonolysis of Sqe in the first flowtube reactor, and (b) their amination products (adducts of SOZ + EA are in red and dehydration products are in blue) upon exposure to ethylamine (EA) in the secondary flowtube reactor.

***Q3****. Conclusion part can be merged into one paragraph.*

    **Reply3**: We have merged them into one paragraph.

***Q4****. Line 324, change "conducted" to "developed", change "were responsible for" to "was responsible for".*

    **Reply4**: We have corrected them.